# Deep versus Handcrafted Tensor Radiomics Features: Prediction of Survival in Head and Neck Cancer Using Machine Learning and Fusion Techniques

**DOI:** 10.3390/diagnostics13101696

**Published:** 2023-05-11

**Authors:** Mohammad R. Salmanpour, Seyed Masoud Rezaeijo, Mahdi Hosseinzadeh, Arman Rahmim

**Affiliations:** 1Department of Integrative Oncology, BC Cancer Research Institute, Vancouver, BC V5Z 1L3, Canada; 2Technological Virtual Collaboration (TECVICO CORP.), Vancouver, BC V5E 3J7, Canada; 3Department of Medical Physics, Faculty of Medicine, Ahvaz Jundishapur University of Medical Sciences, Ahvaz 6135715794, Iran; 4Department of Electrical & Computer Engineering, University of Tarbiat Modares, Tehran 14115111, Iran; 5Department of Physics & Astronomy, University of British Columbia, Vancouver, BC V6T 1Z4, Canada

**Keywords:** head and neck squamous cell carcinomas, deep learning features, radiomic features, hybrid machine learning methods, deep learning algorithms, progression-free survival

## Abstract

Background: Although handcrafted radiomics features (RF) are commonly extracted via radiomics software, employing deep features (DF) extracted from deep learning (DL) algorithms merits significant investigation. Moreover, a “tensor’’ radiomics paradigm where various flavours of a given feature are generated and explored can provide added value. We aimed to employ conventional and tensor DFs, and compare their outcome prediction performance to conventional and tensor RFs. Methods: 408 patients with head and neck cancer were selected from TCIA. PET images were first registered to CT, enhanced, normalized, and cropped. We employed 15 image-level fusion techniques (e.g., dual tree complex wavelet transform (DTCWT)) to combine PET and CT images. Subsequently, 215 RFs were extracted from each tumor in 17 images (or flavours) including CT only, PET only, and 15 fused PET-CT images through the standardized-SERA radiomics software. Furthermore, a 3 dimensional autoencoder was used to extract DFs. To predict the binary progression-free-survival-outcome, first, an end-to-end CNN algorithm was employed. Subsequently, we applied conventional and tensor DFs vs. RFs as extracted from each image to three sole classifiers, namely multilayer perceptron (MLP), random-forest, and logistic regression (LR), linked with dimension reduction algorithms. Results: DTCWT fusion linked with CNN resulted in accuracies of 75.6 ± 7.0% and 63.4 ± 6.7% in five-fold cross-validation and external-nested-testing, respectively. For the tensor RF-framework, polynomial transform algorithms + analysis of variance feature selector (ANOVA) + LR enabled 76.67 ± 3.3% and 70.6 ± 6.7% in the mentioned tests. For the tensor DF framework, PCA + ANOVA + MLP arrived at 87.0 ± 3.5% and 85.3 ± 5.2% in both tests. Conclusions: This study showed that tensor DF combined with proper machine learning approaches enhanced survival prediction performance compared to conventional DF, tensor and conventional RF, and end-to-end CNN frameworks.

## 1. Introduction

Head and neck squamous cell carcinomas (HNSCC) consist of 90% of head and neck (H&N) cancers [1]. About 890,000 new cases of HNSCC are annually diagnosed, and the rates are currently increasing. Moreover, H&N cancer is the seventh most commonly diagnosed cancer worldwide and refers to various malignancies arising from the anatomic sites [2]. HNSCC displays a heterogeneous response to radio-chemotherapy with loco-regional control as well as an overall survival range from 50% to 80% in 5 years [3]. Biomarkers related to HNSCC response to therapy are already known for these types of tumors, such as human papillomavirus infection, epidermal growth factor receptor overexpression, and tumor hypoxia [4]. Although adults younger than 45 years have grown a higher number of HPV, the average age of 60 is estimated for the diagnosis of HNSCC [5].

Medical images such as PET, SPECT, MRI, CT, and others are all employed to provide helpful information about the shape, size, location, and metabolism of H&N cancers [6]. Recent radiomics studies based on CT/PET or fusion of images have been investigated to predict outcomes in patients with H&N cancer [7,8,9,10,11,12]. Image fusion is employed to combine different images generated from different scanners [13]. Clinical outcome prediction is currently challenging and is helpful for treatment planning [14]. Moreover, survival outcome prediction, a regression task [15], provides important prognostic information required for treatment planning [5,7,16,17,18]. Nevertheless, compared to usual regression tasks such as clinical score prediction, clinical outcome prediction is more challenging because survival data is often censored, which means the time of events occurring is vague for some patients [19].

Many studies [7,16,19,20] have focused on prediction of survival outcomes and stages through handcrafted radiomics features (RF). Recently it was suggested [21] that multi-flavoured (tensor) RFs can be invoked to enhance prediction performances. A tensor radiomics paradigm refers to generation of various flavours of a given feature; e.g., from different histogram bin sizes, segmentations, pre-processing filters, fused images, etc. In the present work, we set to predict the survival outcome using tensor deep features (DF) generated through deep learning (DL) algorithms in addition to handcrafted tensor RFs. RFs, such as intensity characteristics, morphological, textural features, and others, enable high-dimensional information extraction from images [22].

Although handcrafted radiomics software (e.g., standardized software [23]) enables RF extraction from the regions of interest, utilizing deep learning (DL) algorithms such as autoencoders enables additional extraction of DFs from images [20]. RFs are a major frontier in medical image analysis, in particular for clinical oncology, since its first introduction by Lambin et al. [24] Radiomics is defined as “the conversion of images to higher dimensional data and the subsequent mining of these data for improved decision support”. RFs have been shown to increase diagnosis accuracy, clinical diagnoses, and prediction of treatment outcomes, bridging between medical imaging procedures and personalized medicine [20,25,26,27]. Although RF analyses are becoming increasingly mature, there are several significant technical limitations, and many RFs, increasingly extracted via standardized radiomics software packages for more reproducible research, are powerless to substantial variations based on image acquisition, reconstruction, and processing procedures; therefore, different feature-generation hyperparameters and segmentation methods may still produce variable RFs [21,28,29].

Moreover, a recent study [30] examined the RF sensitivity to noise, resolution, and tumour volume in the context of a co-clinical trial. This study indicated (i) features from a grey-level run-length matrix, and grey-level size zone matrix were most sensitive to noise, (ii) RF Kurtosis and run-length variance from GLSZM were most sensitive to changes in resolution in both T1w and T2w MRI, (iii), 3D RFs were more robust compared to 2D measures. Another recent study [31] focused on the heterogeneity of RFs by patient-derived tumor xenografts to predict response to therapy in subtype-matched negative triple-negative breast cancer. According to this study, 64 out of 131 preclinical imaging features were identified as being robust.

Furthermore, the encoder layers extract important representations of the image using the feature learning capabilities of the convolutional neural networks. As such, application to different tasks using DFs extracted from DL algorithms has the potential to outperform other imaging features such as RFs, which we investigate in the present work. In fact, autoencoders, such as classification, captioning, and unsupervised learning algorithms, have been explored in the past to use such deep informative features for different predictions or clustering [32].

In the present work, as elaborated next, we first registered PET to CT, cropped, normalized, and enhanced. Subsequently, we extracted handcrafted RFs from 17 images, including fused images and sole CT and PET images using the standardized SERA package. Next, we utilized a DL algorithm to extract DFs from the preprocessed images. Subsequently, a novel “tensor-deep” paradigm was performed to enhance the prediction performance. In tensor DF and RF frameworks, we applied different hybrid systems including dimension reduction algorithms linked with classifiers to predict the survival outcome. Overall, we aimed to understand if and how the use of deep features can add value relative to the use of conventional hand-crafted RFs.

## 2. Methods and Materials

Machine learning algorithms allow improved task performance without being explicitly programmed [33]. Approaches based on machine learning aim to build classification or prediction algorithms automatically by capturing statistically robust patterns present in the analyzed data. Most predictor algorithms are not able to work with a large number of input features, and thus it is necessary to select the optimal few features to be used as inputs. Furthermore, using only the most relevant features may improve prediction accuracy. In what follows, we thus investigate a range of methods for data selection/generation, machine learning, and analysis. In brief, we generated handcrafted RFs as well as DFs and utilized hybrid ML systems (HMLS), including fusion techniques, and dimension reduction algorithms followed by classifiers to enhance classification performance, as shown in Table 1. We develop a tensor DF and RF feature-based prediction framework for patients with H&N cancer from CT, PET, and fused images.

### 2.1. Dataset and Image Pre-Processing

We used a dataset from The Cancer Imaging Archive (TCIA) that included 408 patients with HN cancer (TCIA) who had CT, PET, and clinical data. There were 320 men and 88 women in the dataset. The average age for men was 61.1 ± 11.2 years, and the average age for women was 62.7 ± 9.2 years. Although some studies [33,34,35] focused on the automatic segmentation of regions of interest, this study employed some collaborative physicians to segment the regions of H&N lesions delineated on CT and PET images. The binary progression-free survival outcome includes two classes such as (i) class 0: alive and (ii) class 1: dead. In the pre-processing step, PET images were first registered to CT, and then both images were enhanced and normalized. The bounding box (equal to 224 × 224 × 224 mm^3^) is applied to the images to reduce computation time. We employed 15 fusion techniques shown in Table 1 to combine CT and PET information.

### 2.2. Fusion Techniques

In this study, we employed 15 typical image-level fusion techniques [16,17,21] to combine CT and PET information to combine CT and PET images. Fusion algorithms are shown in Table 1. Figure 1 shows some fused images. Some algorithms such as wavelet fusion, weighted fusion, PCA, and HSI were performed in Python 3.9, and the remaining algorithms were performed in MATLAB 2021b.

### 2.3. Handcrafted Features

In this work, we extracted 215 quantitative RFs from each tumor in 17 images (or flavours), including CT-only, PET-only, and 15 fused CT-PET images through the standardized SERA radiomics software [36]. SERA has been extensively standardized in reference to the Image Biomarker Standardization Initiative (ISBI) [23] and studied in multi-center radiomics standardization publications by the IBSI [32] and the Quantitative Imaging Network (QIN) [37]. There is a total of 487 standardized RFs in SERA, including 79 first-order features (morphology, statistical, histogram, and intensity-histogram features), 272 higher-order 2D features, and 136 3D features. We included all 79 first-order features and 136 3D features [36,37,38].

### 2.4. Deep Features

In the DFs extraction framework, a 3D autoencoder neural network architecture [32] was used to extract DFs thoroughly. Generally, every autoencoder mainly comprises an encoder network and a decoder network. The encoding layer maps the input images to a latent representation or bottleneck, and the decoding layer maps this representation to the original images. So, the number of neurons in the input and output layers must be the same in an autoencoder. Also, the training label is the same as the input data. The encoder follows typical convolutional network architecture, as shown in Figure 2. It consists of three 3 × 3 convolutional layers, each followed by a leaky rectified linear unit (LeakyReLU) and a 2 × 2 max-pooling operation. The pooling layers are used to reduce parameters. The decoder path consists of three 3 × 3 convolutional layers, followed by a LeakyReLU and an up-sampling operation. We used a common loss function for the proposed autoencoder called binary cross-entropy. Thus, the proposed autoencoder was trained with a gradient-based optimization algorithm, namely Adam, to minimize the loss function. We separately applied 17 typical images (CT, PET, and 15 fused images) to the 3D autoencoder model and extracted 15,680 features from the bottleneck layer. This algorithm was performed in Python 3.9.

### 2.5. Tensor Radiomics Paradigm

Figure 3 shows our study procedure. In the image preprocessing step, PET images were first registered to CT, and then both images were enhanced, normalized, and cropped. Next, we employed 15 image-level fusion techniques shown in Table 1 to combine PET and CT images. First, we directly applied CT only, PET only, and 15 fused images to a CNN. Table 1 shows different methods employed in this study.

Subsequently, 215 handcrafted RFs were extracted from each tumor in 17 images (or flavours) including CT-only, PET-only, and 15 fused CT-PET images through the standardized SERA radiomics software. This results in a so-called “tensor radiomics” paradigm where various flavours of a given feature are generated. We recently proposed and studied such a paradigm [21], in which multiple versions (or flavours) of the same handcrafted RFs were generated by varying parameters. The focus of that work was on handcrafted tensor RFs, which we further investigate in this work, and further expand to include deep tensor imaging features. Furthermore, a 3D auto-encoder is used to extract 15,680 DFs. Moreover, a range of optimal ML techniques was selected among different families of classification and dimension reduction algorithms, as could be followed in Appendix A. We then apply DFs vs. handcrafted RFs extracted from each typical image to three sole classifiers such as MLP [39], RFC [40], and LR [41] optimized through five-fold cross-validation and grid-search.

In the tensor DF framework, we first incorporate all features with 17 flavours, remove low variance flavours in the sole dataset via the data preprocessing step, and combine the remaining features via PCA [42]. Subsequently, we employ ANOVA [43] to select the most relevant attributes generated from PCA. Finally, the relevant attributes are applied to three classifiers to predict the survival outcome. For the tensor RF framework, we similarly incorporate all features with 17 flavours, remove the highly correlated features, apply polynomial transform algorithms [44] to the remaining features to combine the flavours, and employ the three mentioned classifiers linked with ANOVA to predict the outcome. In five-fold cross-validation, data points are divided 4-folds for training and 1-fold for testing. Moreover, 80% of training data points were utilized to train the model, and the remaining 20% were utilized to validate and select the best model. Furthermore, the remaining fold was used for external-nested testing. For more consistency, each fold was trained with the same nine classifiers with different optimized parameters, and ensemble-voting (EV) was performed for validation and external testing.

## 3. Results

Following the image preprocessing step, including cropping, normalizing, and enhancing, we employed 15 fusion techniques to combine CT and PET information. Preprocessed sole CT, sole PET, and 15 fused images were directly applied to CNN to predict survival. The highest five-fold cross-validation of 75.2 ± 7% resulted from the DTCWT fusion linked with ensemble CNN. The external nested testing of 63.4 ± 6.7% confirmed the finding. Figure 4 shows the performances which result from CNN.

In the RF framework, we first applied handcrafted RFs extracted from each image to the three above classifiers. As shown in Figure 5, the highest five-fold cross-validation of 70 ± 4.2% resulted from the PCA fusion linked with ensemble RFC. The external nested testing of 67.7 ± 4.9% confirmed the finding, as indicated in Figure 6. Further, the result is not significantly different from the best performance received from the previous section (*p*-values = 0.37 using the paired *t*-test). Moreover, using tensor RFs added value to this study so that the highest five-fold cross-validation of 76.7 ± 3.3% was achieved by the fused tensor RF (FTR) followed by ensemble LR. Moreover, the external nested testing of 70.6 ± 6.7% validated the finding. Overall, by switching from sole CNN to FTR framework, significantly higher performance was achieved (*p*-values < 0.05 using the paired *t*-test).

In the DF framework, we similarly applied DFs extracted from each image to all mentioned classifiers. As shown in Figure 7, the highest five-fold cross-validation of 69.7 ± 3.6% resulted from the SR fusion technique linked with ensemble LR. The external nested testing of 67.9 ± 4.4% validated the finding, as shown in Figure 8. Moreover, there is no significant difference between results arriving from sole CNN and conventional sole DF framework (*p*-values = 0.34 using the paired *t*-test). As can be seen, the highest five-fold cross-validation of 87 ± 3.5% was achieved by the fused tensor DFs (FTD) linked with ensemble MLP. Moreover, the external nested testing of 85.3 ± 5.2% confirmed the finding. Overall, by switching from sole CNN to fused tensor DF (FTD) framework, significantly higher performance was obtained (*p*-values << 0.05 using paired *t*-test). Furthermore, using the FTD framework significantly added value to the survival prediction compared to the FTR framework (*p*-values < 0.05 using the paired *t*-test).

## 4. Discussion

The ability to predict the period of the disease and the impact of interventions is essential to effective medical practice and healthcare management. Hence, accurately predicting a disease diagnosis can help physicians make more informed clinical decisions on treatment approaches in clinical practice [45,46]. The prediction of outcomes using quantitative image biomarkers from medical images (i.e., handcrafted RFs and DFs) has shown tremendous potential to personalize patient care in the context of H&N tumors [47].

The head and neck tumor segmentation and outcome prediction from CT/PET images (HECKTOR) challenge in 2021 [45] aimed at identifying the best approaches to leverage the rich bi-modal information in the context of H&N tumors for the segmentation and outcome prediction so that task 2 of the challenge was defined to predict the progression-free survival. Our previous study [35] investigated the prediction of survival through handcrafted RFs applied to multiple HMLSs, including multiple dimensionality reduction algorithms linked with eight survival prediction algorithms. Thus, the best performance of 68% was obtained with an ensemble voting on these HMLSs.

In the first stage of our study, we applied CNN to CT, PET, and 15 fused images. The highest external nested testing performance of 75.2 ± 7% through DTCWT followed by CNN. Although RFs are the new frontiers of medical imaging processing, using the DF in the prediction of survival can be worth investigating. In this paper, handcrafted RFs were extracted from each region of interest in 17 typical images (or flavours), including CT-only, PET-only, and 15 fused CT-PET images through the standardized SERA radiomics package. A 3D auto-encoder neural network is used to extract 15,680 DFs from all typical images.

Employing conventional handcrafted RFs enabled the highest external nested performance of 67.7 ± 4.9% using the PCA fusion technique linked with RFC. In contrast, using the tensor RF paradigm overcame the usage of conventional handcrafted RFs, resulting in the external nested performance of 70.6 ± 6.7% through ensemble MLP. Therefore, we indicated a significant improvement through FTR (*p*-values < 0.05 using paired *t*-test).

In addition, using sole conventional DFs extracted from each image enabled the highest external nested testing performance of 67.9 ± 4.4% through SR linked with ensemble LR. Meanwhile, in the tensor DF framework, the highest nested performance of 85.3 ± 5.2% was provided from FTDs followed by the ensemble MLP. As shown, our findings revealed that tensor DFs, beyond tensor RF, combined with appropriate ML and employing Ensemble MLP linked with PCA and ANOVA, significantly improved survival prediction performances (*p*-values << 0.05 using paired *t*-test).

Hu et al. [48] developed a DL model that integrates radiomics analysis in a feature fusion workflow for glioblastoma post-resection survival prediction. In this study, the RFs were extracted from the tumor subregions via MR images, including a pre-surgery mpMRI exam with four scans, including T1W, contrast-enhanced T1 (T1ce), T2W, and FLAIR. Moreover, DFs, generated from encoding neural network architecture, in addition to handcrafted RFs enabled the performance of 75%. Furthermore, Lao et al. [49] investigated survival prediction for patients with Glioblastoma Multiforme, who had preoperative multi-modality MR images, through 1403 RFs and 98,304 DFs extracted from transfer learning. Thus, they demonstrated that DFs added value to the prediction task. Since clinical outcome prediction is more challenging because survival data is often censored or the time of events occurring is vague for some patients, survival outcome prediction in this study can provide important prognostic information required for treatment planning.

As a limitation of this study, future studies with a large sample size are suggested. This study considered survival prediction for two-images CT and PET; hence, the proposed approaches can also be used for other related tasks in medical image analysis such as MR images, including T2 weighted image (T2W), diffusion-weighted magnetic resonance imaging (DWI), apparent diffusion coefficient (ADC), and dynamic contrast-enhanced magnetic resonance imaging (DCE-MRI). For future research, it is suggested that relevant evaluations be carried out based on DFs and RFs with larger data sets and other anatomical areas. Another crucial step towards a reproducible study is the use of robust RFs or DFs to analyze variations as hundreds of feature sets are extracted from medical images. In *future* study, we thus aim to study reproducible imaging features to predict overall survival outcomes.

The novelty of this study is the usage of different fusion techniques to generate new images from CT and PET images, instead of using different scanners, to construct a tensor paradigm. Further, employing DL algorithms to generate DFs boosted the tensor paradigm so that a significant improvement compared to the TR paradigm or DL algorithms was achieved. In addition, using the same nine classifiers with different optimized parameters enabled us to do ensemble voting for consistent results.

## 5. Conclusions

This study demonstrated that the use of a tensor radiomics paradigm, beyond the conventional single-flavour paradigm, enhances survival prediction performance in H&N cancer. Moreover, tensor DFs significantly surpassed tensor RFs in the prediction of outcome. We obtained the highest external nested testing performance of 85.3 ± 5.2% for survival prediction through fused tensor DF (FTD) followed by an ensemble MLP technique. Moreover, this approach outperformed employing end-to-end CNN prediction. In summary, to significantly enhance survival prediction performance, quantitative imaging analysis, especially usage of tensor DFs, can be valuable.

## Figures and Tables

**Figure 1 diagnostics-13-01696-f001:**
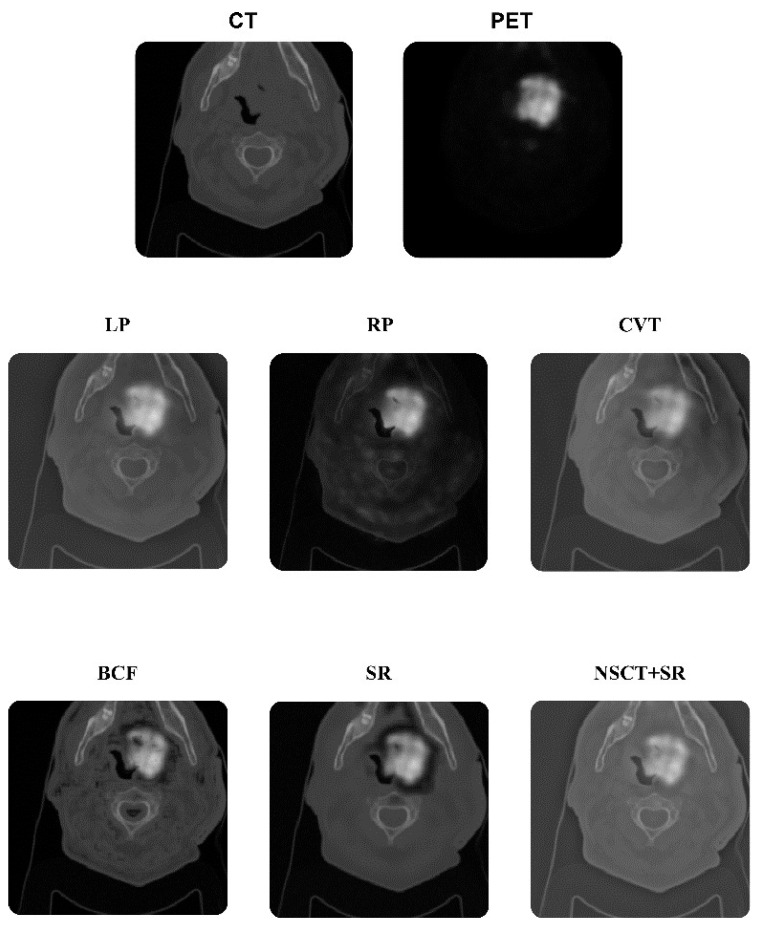
Fusion process. Some examples of (1) CT image alone, (2) PET image alone, (3) fusion of laplacian pyramid (LP), (4) ratio of the low-pass pyramid (RP), (5) curvelet transform (CVT), (6) sparse representation (SR), (7) nonsubsampled contourlet transform (NSCT) + SR, (8) bilateral cross filter (BCF).

**Figure 2 diagnostics-13-01696-f002:**
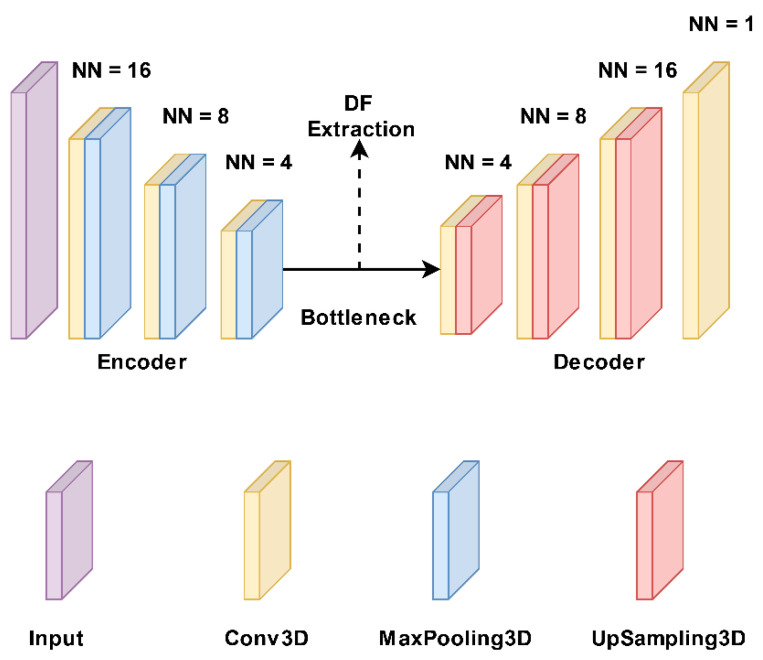
Structure of our autoencoder model. It includes three 3 × 3 convolutional layers, each followed by a leaky rectified linear unit (LeakyReLU) and a 2 × 2 max-pooling operation. The decoder path includes three 3 × 3 convolutional layers, followed by a LeakyReLU and an up-sampling operation. NN: number of neurons.

**Figure 3 diagnostics-13-01696-f003:**
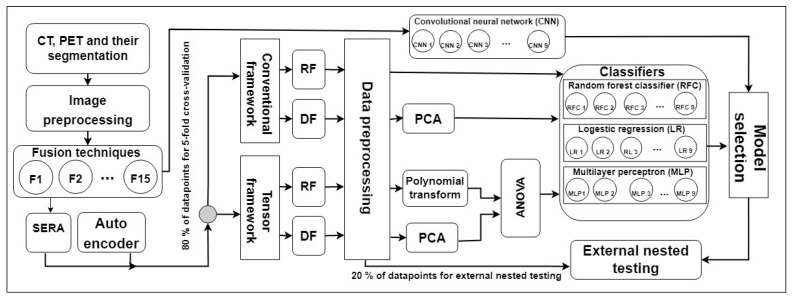
Shows our study procedure. F: fusion technique, RF: radiomics feature, DF: deep feature, PCA: principal component analysis, CNN: convolutional neural network, RFC: random forest classifier, MLP: multilayer perceptron, LR: logistic regression, and ANOVA: analysis of variance feature selector.

**Figure 4 diagnostics-13-01696-f004:**
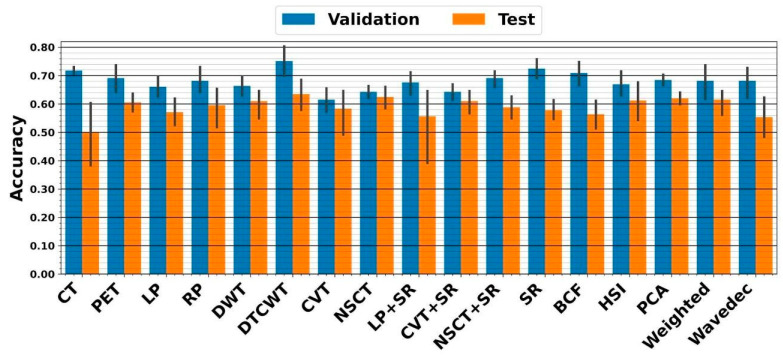
Performance from five-fold cross-validation and external nested testing. The X-axis indicates CT, PET, and 15 fused images.

**Figure 5 diagnostics-13-01696-f005:**
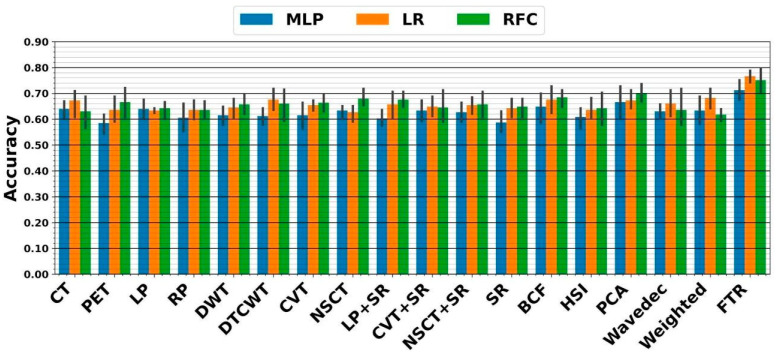
Performance from five-fold cross-validation in the RF framework. The X-axis indicates RFs extracted from CT, PET, and 15 fused images, as well as the proposed fused tensor RF (FTR) approach.

**Figure 6 diagnostics-13-01696-f006:**
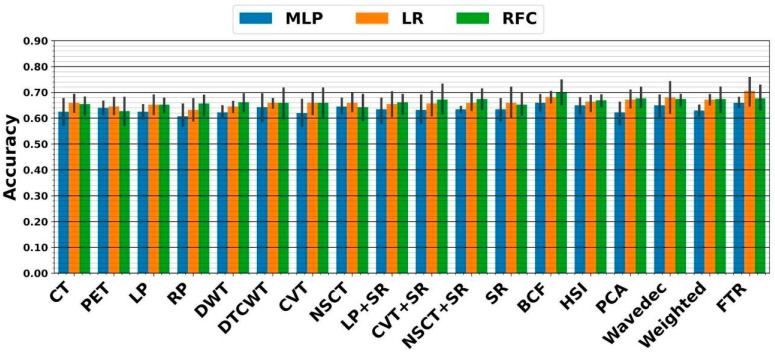
Performance from external nested testing in the RF framework. The X-axis indicates RFs extracted from CT, PET, and 15 fused images, as well as the proposed fused tensor RF (FTR) approach.

**Figure 7 diagnostics-13-01696-f007:**
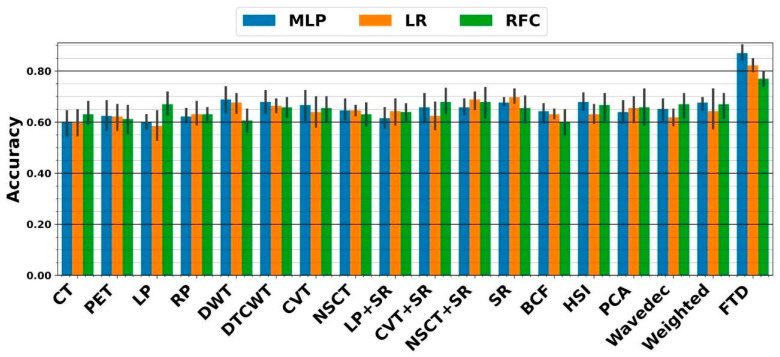
Performance from five-fold cross-validation in the DF framework. The X-axis indicates DFs extracted from CT, PET, and 15 fused images, as well as the proposed fused tensor DF (FTD) approach.

**Figure 8 diagnostics-13-01696-f008:**
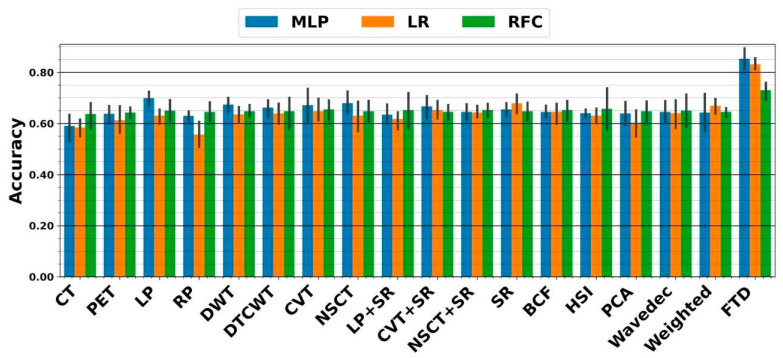
Accuracies achieved from external nested testing in the DF framework. The X-axis indicates DFs extracted from CT, PET, and 15 fused images, as well as the proposed fused tensor DF (FTD) approach.

**Table 1 diagnostics-13-01696-t001:** List of the employed algorithms.

Category	Algorithms
Fusion techniques	*Laplacian Pyramid (LP), Ratio of the low-pass Pyramid (RP), Discrete Wavelet Transform (DWT), Dual-Tree Complex Wavelet Transform (DTCWT), Curvelet Transform (CVT), NonSubsampled Contourlet Transform (NSCT), Sparse Representation (SR), DTCWT + SR, CVT + SR, NSCT + SR, Bilateral Cross Filter (BCF), Wavelet Fusion, Weighted Fusion, Principal Component Analysis* *(PCA), and Hue, Saturation and Intensity fusion (HSI)*
Dimension reduction algorithms	*Analysis of Variance (ANOVA) and Principal Component* *Analysis (PCA)*
Classifiers	*Multilayer Perceptron (MLP), Random Forest, and Logistic Regression (LR)*

## Data Availability

All datasets were collaboratively pre-processed by Qurit Lab (Quantitative Radiomolecular Imaging and Therapy, qurit.ca, accessed on 8 March 2023) and the Technological Virtual Collaboration Corporation Company (TECVICO CORP., tecvico.com, accessed on 8 March 2023). All Codes were also developed by both collaboratively. All codes (including predictor algorithms, feature selection algorithms, and feature extraction algorithms) and datasets are publicly shared at: https://github.com/Tecvico/Tensor_deep_features_Vs_Radiomics_features, accessed on 8 March 2023.

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
