# Peer review of "Deep versus Handcrafted Tensor Radiomics Features: Prediction of Survival in Head and Neck Cancer Using Machine Learning and Fusion Techniques"

_diagnostics, 2023, doi:10.3390/diagnostics13101696_

Round 1

Reviewer 1 Report

To predict the survival outcome in head and neck cancer, this paper aims to report performance differences of deep features (DFs) against handcrafted features under conventional and tensor frameworks. Also, reporting performance using fusion-based technique. Results demonstrated
good results of models utilizing the tensor-based framework performed better than those of the conventional framework. Also, fused-based techniques generated good results when coupled with the DFs of the tensor framework. The paper is well-motivated and informative in terms of reporting the performance differences and to identify
the best-performing approach. However, the following needs to be addressed:

-For survival outcomes, specify the binary outcomes (i.e., possible class labels). Better to include a problem statement section

-Include other machine learning algorithms such as SVM and XGBOOST

-Include a Table to briefly summarize methods

-Report results using performance measures such as AUC, F1, and Matthews correlation coefficient.

-For each model,  report the combined confusion matrices of five-fold cross-validation (when applied to testing in each iteration).

-In "2.1 Dataset and Image Pre-Processing" section,
it is stated that "We employed 15 fusion techniques mentioned in 2.2.1". However. 2.2.1 is incorrectly references. Kindly
fix it

-In 2.5. Tensor Radiomics Paradigm",
it is stated that "15 image-level fusion techniques mentioned in 2.3.1 to combine PET and CT images"
However, 2.3.1 is not correctly references (i.e., not available). Needs to be fixed

-Presentation needs to be improved

Author Response

Thank you for your valuable time to review our work. We hope we have properly addressed your comments, which we discuss in the below attachment. We remain open to further feedback.

Reviewer 2 Report

This study demonstrated that the use of the tensor paradigm, beyond the conventional single-flavor paradigm, enhances survival prediction performance in H&N cancer. Moreover, tensor DFs significantly surpassed tensor RFs in the prediction of outcome. Thus, we obtained the highest external nested testing performance of 85.3% ± 5.2% for survival prediction through fused tensor DF (FTD) followed by an ensemble MLP technique. Moreover, this approach outperformed employing end-to-end CNN prediction. In summary, to significantly enhance survival prediction performance, quantitative imaging analysis, especially the usage of tensor DFs, can be valuable.

The paper is interesting. I have following concerns, the authors need to address the following.

* The abstract should be an objective representation of the article: it must not contain results which are not presented and substantiated in the main text and should not exaggerate the main conclusions.

* In the introduction, what key theoretical perspectives and empirical findings in the main literature have already informed the problem formulation? What major, unaddressed puzzle, controversy, or paradox does this research address? * Why does it need to be addressed?

* Why it should be now - not in the past?

* Further, in the introduction, what is the recent knowledge gap of the main literature that the author needs to write this research? What we have known and what we have not known? What is missing from current works? Please explain and give examples!

* In terms of the knowledge gap, it will be best if the research challenge/knowledge gap could be stated in one article or more articles in the main literature. Assure that you have included all key articles (e.g., most widely cited articles) in the main literature. Mention them. 1) Roy, Sudipta, Timothy D. Whitehead, James D. Quirk, Amber Salter, Foluso O. Ademuyiwa, Shunqiang Li, Hongyu An, and Kooresh I. Shoghi. "Optimal co-clinical radiomics: Sensitivity of radiomic features to tumour volume, image noise and resolution in co-clinical T1-weighted and T2-weighted magnetic resonance imaging." EBioMedicine 59 (2020): 102963.    2) Roy, Sudipta, Timothy D. Whitehead, Shunqiang Li, Foluso O. Ademuyiwa, Richard L. Wahl, Farrokh Dehdashti, and Kooresh I. Shoghi. "Co-clinical FDG-PET radiomic signature in predicting response to neoadjuvant chemotherapy in triple-negative breast cancer." European Journal of Nuclear Medicine and Molecular Imaging (2022): 1-13.    3)  Roy, Sudipta, Tanushree Meena, and Se-Jung Lim. "Demystifying supervised learning in healthcare 4.0: A new reality of transforming diagnostic medicine." Diagnostics 12, no. 10 (2022): 2549.   the impact of traditional level set with radiomics need to be discussed: 4) Roy, Sudipta, Debnath Bhattacharyya, Samir Kumar Bandyopadhyay, and Tai-Hoon Kim. "An iterative implementation of level set for precise segmentation of brain tissues and abnormality detection from MR images." IETE Journal of Research 63, no. 6 (2017): 769-783.

* Practical Implications seem to be unclear? Please mention and make a reference!  As to practical implications, how do the findings help the health organizations? Please explain and give examples! Assure that any recommendation is clear and actionable for organizations.

* The sensitivity of radiomics is ignored. Need to focus more on the sensitivity and reproduciablity and standarization. 

* What types of deep learning / traditional models are used ? and why ?

* What are the challenges and opportunities in radiomics?

Author Response

(The authors gave the same response as above.)

Round 2

Reviewer 1 Report

Authors addressed comments raised in a previous round of review.

Minor comment:

In "Confusion_matrix and other evaluation measurments.xlsx" file, include

additional rows averaging performance results of each 5 folds

Author Response

Thank you for your valuable time to review our work. We hope we have properly addressed your comment. We have updated the excel file shared in the GitHub page and added some rows indicating average of folds (performance) and its standard deviation. 

You can follow it through this link: 

https://github.com/Tecvico/Tensor_deep_features_Vs_Radiomics_features/tree/main/Results

Reviewer 2 Report

Authors have revised the paper as per the suggestion. I have no further comments for this paper.

Author Response

Sounds great. Thank you for your valuable time to review our work.